# Role and Modulation of TRPV1 in Mammalian Spermatozoa: An Updated Review

**DOI:** 10.3390/ijms22094306

**Published:** 2021-04-21

**Authors:** Marina Ramal-Sanchez, Nicola Bernabò, Luca Valbonetti, Costanza Cimini, Angela Taraschi, Giulia Capacchietti, Juliana Machado-Simoes, Barbara Barboni

**Affiliations:** 1Faculty of Biosciences and Technology for Food, Agriculture and Environment, University of Teramo, 64100 Teramo, Italy; nbernabo@unite.it (N.B.); lvalbonetti@unite.it (L.V.); ccimini@unite.it (C.C.); ataraschi@unite.it (A.T.); gcapacchietti@unite.it (G.C.); jsmachadosimoes@unite.it (J.M.-S.); bbarboni@unite.it (B.B.); 2Institute of Biochemistry and Cell Biology (CNR-IBBC/EMMA/Infrafrontier/IMPC), National Research Council, Monterotondo Scalo, 00015 Rome, Italy; 3Istituto Zooprofilattico Sperimentale dell’Abruzzo e del Molise “G. Caporale”, Via Campo Boario 1, 64100 Teramo, Italy

**Keywords:** TRPV1, spermatozoa, thermotaxis, chemotaxis, endocannabinoid system, mammals, human, sperm signaling, ion channel

## Abstract

Based on the abundance of scientific publications, the polymodal sensor TRPV1 is known as one of the most studied proteins within the TRP channel family. This receptor has been found in numerous cell types from different species as well as in spermatozoa. The present review is focused on analyzing the role played by this important channel in the post-ejaculatory life of spermatozoa, where it has been described to be involved in events such as capacitation, acrosome reaction, calcium trafficking, sperm migration, and fertilization. By performing an exhaustive bibliographic search, this review gathers, for the first time, all the modulators of the TRPV1 function that, to our knowledge, were described to date in different species and cell types. Moreover, all those modulators with a relationship with the reproductive process, either found in the female tract, seminal plasma, or spermatozoa, are presented here. Since the sperm migration through the female reproductive tract is one of the most intriguing and less understood events of the fertilization process, in the present work, chemotaxis, thermotaxis, and rheotaxis guiding mechanisms and their relationship with TRPV1 receptor are deeply analyzed, hypothesizing its (in)direct participation during the sperm migration. Last, TRPV1 is presented as a pharmacological target, with a special focus on humans and some pathologies in mammals strictly related to the male reproductive system.

## 1. Introduction

Transient receptor potential vanilloid 1 (TRPV1) is one of the most studied members within the big family of transient receptor potential (TRP) channels. The structure and function of this protein have been widely studied by numerous researchers in order to decipher its exact mechanism of action, although some aspects remain unknown. TRPV1 has been found in multiple cell types, organs, and tissues from different species, and a great number of articles and reviews have investigated and discussed this polymodal channel [1,2,3,4]. TRPV1 acts as a multimodal sensor for numerous physical, chemical, and mechanical studies by being a key player in signal transduction in multiple processes.

In the present review, we focused on the role of TRPV1 on spermatozoa physiology after ejaculation. This receptor has been shown to participate in multiple events of the sperm functions, such as the acrosome reaction, calcium trafficking, and fertilization. Here, based on exhaustive bibliographic research of international peer-reviewed literature (literature found in the database Scopus), TRPV1, its subfamily vanilloid TRP, and the big TRP family are presented and discussed. The protein structure and its multiple modulators found in different cell types and species are reviewed as well. After examining the modulators of TRPV1 previously described, only the ones found in the reproductive tract, seminal plasma, or spermatozoa with a potential role in sperm physiology were considered for further investigation in this review.

One of the most intriguing events of the post-ejaculatory life of the spermatozoa in the migration through the female tract and toward the fertilization site. During this trip, which may last from hours to days (depending on the species) [5,6], spermatozoa reside within the oviduct, where they remain bound to the oviductal epithelium [7] or are guided by various stimuli, such as rheotaxis (caused by the peristaltic movements of the oviduct smooth muscle and the fluid production), chemotaxis (promoted by the response to chemical factors) and thermotaxis (induced by the gradient of temperature). These guiding mechanisms and their relationship with the TRPV1 receptor are also examined thoroughly in this review. For instance, this channel protein has been recently recognized as a heat-sensing receptor during thermotaxis in humans [8]. Finally, the use of TRPV1 as a pharmacological target in humans and the incidence of pathologies related to the male reproductive system in mammals are explored.

## 2. TRPV1: Family and Subfamily

TRPV1 has been shown to be present in numerous tissues and cells of different species, from the urinary bladder [9], corneal fibroblasts [10], brain [11], and breast cancer cells [12], to the reproductive system, as female organs and placenta [13,14]. TRPV1 belongs to the vanilloid TRP subfamily, which also comprises TRPV2, TRPV3, TRPV4, TRPV5, and TRPV6. Figure 1 shows a graphical phylogenetic reconstruction of the TRPV subfamily, while Appendix A shows the alignment and phylogenetic reconstruction, performed using the function “build” of ETE3 v3.1.1 [15] as implemented on the GenomeNet (https://www.genome.jp/tools/ete/; accessed on 20 April 2021). Fasta sequences were obtained from UniProtKB (https://www.uniprot.org/; accessed on 20 April 2021). The tree was constructed using fasttree with slow NNI and MLACC=3 (to make the maximum-likelihood NNIs more exhaustive) [16]. Values at nodes are SH-like local support.

In addition to TRPV, six more subfamilies are included within the big TRP family, differentiated by their amino acid sequence and structural homology [17]: canonical TRP channels (TRPC); melastatin TRP channels (TRPM); polycystin TRP channels (TRPP); mucolipin TRP channels; ankyrin TRP channels; and no mechanoreceptor potential C (nompC) TRP (TRPN) channels.

TRPs constitute a group of ion channels that serve as cellular sensors for multiple physical and chemical stimuli and are considered the most diversified group of ion channels [18]. They are related to signal transduction and sensitivity to different mechanical, physical, and chemical (exogenous and endogenous) stimuli. Moreover, these receptors are known to be very versatile and diverse in their regulatory mechanism, gating, and selectivity, being generally characterized as non-selective to cations and thus conferring them the ability to influx several monovalent and divalent cations upon activation, including Ca^2+^ ions [19], K^+^, and Na^+^, as well as cyclic nucleotides [20].

TRPV channels are the best-studied TRP subfamily, probably due to the role played in the pain pathway [21]. While TRPV1 to TRPV4 are Ca^2+^ permeable, non-selective cationic channels, thermosensitive and sensitive to endogenous and exogenous ligands TRPV5 and TRPV6 are relatively more selective for Ca^2+^ ion and not thermosensitive [18,22].

TRPV1 was the first mammalian TRP channel to be cloned in 1997 [21]. Since then, many researchers around the globe have focused their efforts on deciphering the protein structure and the mechanisms regulating its function.

## 3. TRPV1: Protein Structure

The structure of the TRPV1 channel was first solved in 2008 by Moiseenkova-Bell et al. [23]. Similar to other voltage-gated channels, TRPV1 is a six-pass transmembrane protein (S1–S6) exhibiting a four-fold symmetry around a central ion pathway, with a reentrant pore loop (PL) (hydrophobic handle) between the 5th and 6th transmembrane helices (S5 and S6, respectively), an intracellular N-terminus rich in ankyrin repeats [24] and a relative short C-terminal region [1,25,26]. The basic structure is shared with the other members of the TRP channels family, and the variability in the amino acid sequence in the 4th transmembrane helix is responsible for the voltage-dependent activation [1,20]. A representative image of the channel can be observed in Figure 2.

One of the most interesting features of the TRPV1 structure is the existence of intracellular domains (e.g., the N-terminal domain) key for the interaction with other proteins, as well as the presence of binding sites for different molecules able to modulate the protein function. For example, calmodulin (CaM) protein has been found to bind to the first ankyrin repeat of the N-terminal domain [27]. Ankyrin repeats consist of approximately 33 residues repeating motif containing 24 copies of these repeats and linked to the S1 by a 77-amino-acid segment [28]. These domains are known to be present in over 400 proteins with numerous functions, mainly protein-protein interaction, and may also serve as a binding site for molecules such as ATP.

The structure of the C-terminal domain, except for two isolated fragments, is unavailable. However, through experiments of single-particle cryo-electron microscopy, the TRP box placed in this region was shown to be oriented toward the S4–S5 linker and the pre-helix-S1, suggesting its essential role in the allosteric modulation of the protein [26].

Aimed to provide a general vision of the protein structure, Figure 3 shows a three-dimensional image of the TRPV1 structure determined using single-particle electron cryo-microscopy by Liao et coll. [26]. The 3D structure can be viewed at https://www.rcsb.org/structure/3J5P (assessed on 20 April 2021), PDB ID 3J5P [29,30].

## 4. TRPV1: Function, Modulators, and Mechanisms of Regulation

Due to its wide expression in the peripheral and central nervous system (including primary sensory neurons, spinal cord, and brain), the TRPV1 channel has been demonstrated to be involved in mediating the sensation of pain in inflammation, and it was shown during the intense burning sensation after a heat exposure above 42 °C. However, the widespread expression of TRPV1 not only within the nervous system but in numerous tissues and organs in humans and other mammals suggests its involvement in multiple important functions also in non-pain-related physiological and pathological conditions [3]. For instance, TRPV1 has been demonstrated to be an active player in the cell-environment crosstalk influencing cell behavior, such as in cancer and immunity. It has been found to be expressed in cells of the innate and adaptative immune system, although its exact role remains to be determined [31]. Moreover, the involvement of this ion channel in energy homeostasis was recently discovered, probably in the control of food intake and energy expenditure [32].

TRPV1 displays extraordinary flexibility of movement in response to different agonists [33]. Several agonists and modulators (exogenous and endogenous) of the TRPV1 function have been identified in the last 30 years. Figure 4 gathers the most common and well-known modulators of the protein. It is important to note that the classification in endogenous and exogenous depends on the source of the compounds referred to the cell, i.e., endogenous stimuli are substances produced by the cell (e.g., the spermatozoa), while exogenous source include compounds that can be found inside the organism (e.g., in the follicular fluid).

The two classical and widely studied agonists of TRPV1 are capsaicin and resiniferatoxin (this last considered an ultrapotent agonist) [37], although recent experiments have shown faster and longer-lasting analgesic effects by targeting some new compounds such as (not specified by the authors) [90]. The exact mechanism by which capsaicin induces the opening of TRPV1 and then its activation is not completely known yet. However, it has been demonstrated the ability of capsaicin (a lipophilic molecule) to permeate the cell plasma membrane, probably by penetrating the bilayer before flipping from the extracellular to the intracellular leaflet [91]. Then, capsaicin may bind TRPV1 at an unknown site, activating the channel and thus allowing the ATP binding to the ankyrin repeats, modulating the calcium-dependent channel desensitization [92,93].

Despite the efforts made by numerous researchers, to date, very little is known about the exact mechanisms of activation of TRPV1, giving rise to some hypotheses [94] and a consensus on the existence of various opening gates within the channel. Moreover, it is important to consider that, as discussed by Rosenbaum et coll. (2007) [95], binding sites are different from gating sites, which are also able to induce changes in the allosteric properties of the protein, thus sensitizing the channel to other agonists. As explained by the authors, capsaicin can sensitize TRPV1 to protons and heat even at low concentrations, unable to produce a direct activation [95].

What is well-known is the presence of TRPV1 in membrane microdomains (lipid rafts), i.e., cholesterol- and sphingolipid-enriched membrane domains characterized by their insolubility in Triton X-100 detergent (also known as detergent-resistant membranes, DRMs). These rafts play a pivotal role in cell signaling by being the residing site for many proteins embedded within the membrane or acting as a binding or interacting site. Even if TRPV1 concentrations are estimated to be low in normal conditions, Botto et coll. (2010) have demonstrated how bicarbonate exposure induces the protein increase within the lipid microdomains [96], thus exposing the ability of this receptor to translocate in response to certain stimuli and in certain biochemical contexts, hypothesizing an integrated functional dialogue between membrane microdomain architecture and the endocannabinoid system.

## 5. Role of TRPV1 on Spermatozoa

TRPV1 involvement in sperm cells maturation and function has been demonstrated by different researchers in the last years. It has been reported to be present in mature spermatozoa in many species (duck [18], fish [97], bull [98], boar [14], and human [8,99]). TRPV5 has also been detected in rats [100], while TRPV4 may be present in various vertebrates sperm cells. The last has been observed to be present as an N-glycosylated protein, which activation induces a Ca2+ influx, suggesting its involvement in sperm motility [101]. Table 1 gathers all the species where TRPV1 was found on spermatozoa, while Table 2 presents all the modulators from Figure 4 that have been found in the female reproductive tract or in the seminal plasma and on spermatozoa, and which relationship with the process of reproduction has been already described or exit enough evidence to be hypothesized.

TRPV1 has been suggested to have an important physiological role in sperm capacitation and acrosome reaction (AR), thus contributing to the complex process of fertilization [133]. Indeed, in mammals, fertilization involves multiple events, initiating either with the sperm cells entrance into the cervix and the later headway of the healthiest spermatozoa [5,113] toward the uterus or the direct deposition within the uterus, depending on the species. Once the spermatozoa reach the oviduct [134], they bind to the oviductal epithelium for hours to days forming a functional sperm reservoir as demonstrated in several mammalian species [135]. After this indefinite period of time and by a still partially unknown mechanism [7], spermatozoa detach from the oviduct and continue their way toward the oocyte, interacting afterward with the proteins from the zona pellucida (ZP) of the egg and thus triggering the release of the acrosome content with the enzymes necessary to penetrate the ZP. Once the sperm cells adhere to the plasma membrane via its equatorial region, fuse, and incorporate the spermatozoon into the ooplasm until the fertilization is completed. Thus, the zygote can divide by successive mitosis to form an embryo [136].

One of the current challenges in regard to the fertilization process remains the understanding of the complex and still completely unknown communication between the spermatozoa and the female environment, which prevents premature capacitation and helps to preserve the sperm fertilizing potential until the encounter with the egg. To achieve a successful sperm-egg interaction, two conditions are mandatory: both capacitated spermatozoa and mature oocytes must be present at the same location in the oviduct. The first condition implies that spermatozoa should acquire the fertilizing ability through the process of capacitation in view of the meeting with the egg, experiencing a series of complex events that modify some of their biochemical, chemical, or physicochemical features. The second one requires the participation of the intricate neuroendocrine female axis, which by the active role of hormones such as progesterone and estradiol interacts not only with the female tract (for instance, the ovary and oviducts) but also directly with the spermatozoa, contributing to the successful encounter between the male and female gametes. However, numerous mechanisms involved in mammalian fertilization remain to be elucidated, such as the sperm/egg ratio (specifically at the fertilization site) that is somehow and highly regulated to avoid polyspermy.

One of the fertilization events that still merits further investigation in the reproduction field is the sperm selection that is taking place in the female genital tract, from where the semen is deposited to the site of fertilization. This issue has attracted the attention of numerous researchers during the last decades (see the reviews by Suarez and Pacey 2006 [137] and Miller 2018 [138]). Along with the active peristaltic movements due to the contractions of the oviduct smooth muscle and the fluid production (rheotaxis) [139], chemotaxis and thermotaxis have been proposed as the mechanisms that help the spermatozoa to reach the fertilization site once in the oviducts [140]. It is thought that, in vivo, thermotaxis, rheotaxis, and chemotaxis are complementary, involving spermatozoa being first guided by thermotaxis and rheotaxis and then mainly by chemotaxis when they are proximal to the oocyte [141]. However, the most plausible explanation is that the three mechanisms could occur simultaneously in some regions of the oviduct, contributing together to the arrival of the sperm cell to the meeting with the oocyte. It is worthy to note that the oviduct should be intended as a 4D system in which every event occurs in a certain location and time, which could help to understand the variations in the hormone concentrations (e.g., progesterone or estradiol) in a precise location and period of time. Thus, rheotaxis and thermotaxis may act in a first moment, being then not replaced but supported by the chemical stimuli to complete the journey. Figure 5 shows a diagram of the oviduct of mammals, with the three sperm gradients in the tract where these mechanisms are supposed to exert their guidance. TRPV1 acts as a polymodal sensor, reacting in a graded manner to diverse physical, mechanical, and chemical stimuli and changing its response depending on the concentrations, agonists, modulators, and even the environment [142]. Its function within these migration processes is discussed below.

### 5.1. Sperm Chemotaxis, TRPV1, and the Endocannabinoid System (ECS)

As shown by numerous researchers [139,143,144,145,146,147], chemotaxis guides spermatozoa and helps them to find the oocyte due to the great sensitivity of the sperm cells to picomolar concentrations of progesterone (P4) [144] and other molecules from the follicular or oviductal fluid [141]. On this basis, hormones such as P4, secreted from the cumulus cells surrounding the oocyte, act as a sperm chemoattractant [144,148], demonstrated on different animal species as human [144], rabbit [149], equine [147], mouse [150] and swine [146]. P4 then induces the influx of Ca^2+^ by activating progesterone receptors as CatSper receptors [151], participating in the sperm migration. Since extracellular Ca^2+^ is a necessary condition for sperm chemotaxis, the inhibition of the calcium channels may block the process of sperm migration, as demonstrated in some species [152,153].

Interestingly, a multifunctional molecule has been shown to modify the function of numerous ion channels, probably by altering the thickness and elastic properties of the surrounding lipid bilayer [154] or interacting with protein channels [155]. This molecule is cholesterol, a major component of plasma membranes that confers them high mechanical strength and preventing from cell lysis. Their concentrations vary from cell to cell and from one species to another, while the distribution of this amphipathic molecule is confined to the membrane microdomains or lipid rafts. Some researchers have evaluated the role of cholesterol on TRPV1 function. For instance, Picazo-Juarez et coll. (2011) found a cholesterol-binding site in TRPV1, precisely in the S5 helix. The existence of this binding motif has been proposed as responsible for the cholesterol sensitivity of TRPV1 [156]. In addition to that, some researchers have demonstrated a depletion of the TRPV1 protein amount after cholesterol depletion in adult rat dorsal root ganglion neurons [157], suggesting that TRPV1 could be localized in the cholesterol-rich microdomains, while studies in the cell line HEK293 found TRPV1 in cholesterol-poor (non-rafts) microdomains but not in cholesterol-rich rafts [156]. Storti et coll. (2015) suggested a spatial-temporal regulation of TRPV1 dynamics, proposing three populations of TRPV1: the first able to bind caveolin-1 protein (present in the rafts); the second interacting with tubulin; and the third as a freely-fast diffusion fraction [158]. With this information, it is possible to suggest a species- and cell-specific role of cholesterol modulation of the TRPV1 function, with important participation of different proteins from the raft microdomains.

In addition to the physiological role exerted by TRPV1 in the transmembrane calcium trafficking [159], it was also demonstrated to be a molecular target for a great variety of molecules (evidenced in navy blue in Figure 4) that were found either within the oviductal fluid or directly synthesized by the sperm cells. This complex ensemble of molecules in which concentrations are spatial-temporal dependent influence in a coordinated manner the sperm migration through the oviduct.

Among these molecules (evidenced in navy blue in Figure 4), anandamide stands out as an example, a molecule from the endocannabinoid system. Some molecules and receptors from the endocannabinoid system have been demonstrated to be involved in the dialogue between the female oviductal fluid and the spermatozoa during the sperm migration along the reproductive tract. Anandamide (arachidonoylethanolamide, AEA) is synthesized, degraded, bound, and transported by the biochemical machinery present in mammalian spermatozoa [14]. This major endogenous endocannabinoid/endovanilloid ligand is present in the uterus and the implantation site [160,161], where AEA was found to stimulate the migration of endometrial stromal cells via type 1 cannabinoid receptor (CB1R) in a dose-dependent manner by activating the ERK1/2 and PI3K/Akt pathways [162]. It is also present in seminal plasma and uterine fluids [14], activating or inhibiting cannabinoid receptors depending on its concentration, producing a dual-stage effect that could be explained in a three-step process:

First, the high levels of AEA within the seminal plasma and the uterine fluid may prevent the sperm cells from premature capacitation by acting as an inhibitor. AEA interacts with CB1R, a Gi/o protein-coupled receptor in which activation, concurrently with the low levels of bicarbonate, prevents the increase in the membrane fluidity, known as a key feature of capacitated spermatozoa. In fact, during the normal process of capacitation, the plasma membrane (PM) and the outer acrosome membrane (OAM) become less stable and gradually acquire the ability to fuse (fusogenicity) [163,164]. This status is possible due to the high-stable asymmetrical organization of the membranes, in which the aminophospholipids phosphatidylserine (PS) and phosphatidylethanolamine (PE) are concentrated in the inner leaflet, and the choline phospholipids sphingomyelin (SM) and phosphatidylcholine (PC) in the outer leaflet of the sperm head. This asymmetry is established and maintained by the action of several translocating enzymes with various phospholipid specificities and modulated by PKA-dependent phosphorylation. Thus, if fusogenicity increases, AR could take place before the meeting with the ZP of the oocyte. If AR is not fully completed, the spermatozoa are unable to release the acrosome content, preventing precocious capacitation or death of the sperm cell.

In the second step, spermatozoa experience a variation in the AEA levels during their way toward the oocyte, moving from high concentrations of AEA to low concentrations (activating state) within the oviduct [165]. At the same time, the concentration of the ion bicarbonate increases, producing a migration of CB1R to the equatorial region of the sperm and thus attenuation of its activity. It then activates the soluble isoform of adenylyl cyclase (sAC), giving rise to an increase in the intracellular cAMP levels acting via a PKA-dependent pathway [14,99,166]. This increase in the intracellular AMP, along with many others, contributes to the sperm acquisition of their fertilizing ability within the oviduct in vivo or after incubation under capacitating conditions in vitro. The PM phospholipid redistribution and lipid scrambling take place, leading to an increase in the sperm membrane fluidity and disorder and the cholesterol efflux by soluble protein acceptors from the anterior sperm head. The calcium-dependent polymerization of G-actin provokes the formation of an F-actin network that forms a diaphragm between PM and OAM, a structure of great importance for signal transduction [167].

During the third step, an increase in the endogenous AEA concentrations (secreted by the sperm cells) is likely to play a role in stabilizing the acrosome membrane, thus preventing non-targeted AR by interacting this time with TRPV1.

Ref. [14,99], which acts as a controller of the F-actin network. TRPV1, located over the post-equatorial area of the sperm head, is inactive at early capacitation stages. As capacitation progresses, this protein translocates from the post-equatorial to the anterior region of the sperm head, becoming active. Its activation thus triggers a membrane depolarization wave, opening the voltage-operated calcium channels (VOCC), leading to an increase in the intracellular calcium concentration. This calcium gradient is able to activate some receptors, such as TRPV1, which may contribute to the modulation of the capacitation status. The increase in the intracellular calcium concentrations helps then in the F-actin depolymerization and the disappearance of the actin network, allowing the PM and OAM interaction.

Altogether, the information conveyed so far suggests an indirect role for TRPV1 in sperm chemotaxis, which may be controlled by molecules of the endocannabinoid system such as anandamide. Indeed, the concentrations of AEA and its differences of concentration within the female tract could influence sperm migration, as in other cell types. For example, Ca^2+^ influx upon activation of CB1R and TRPV1 by AEA has been demonstrated to induce the sperm release from the oviductal sperm reservoir in bovine [168]. Another study performed in sea urchins by Guerrero et coll. (2010) suggested the sperm’s ability to increase the concentration of Ca^2+^ in the flagellum. Moreover, they observed a curvature in the presence of a negative chemoattractant gradient, which could constitute a characteristic of chemotactic motility not only in sea urchin spermatozoa but in other species as well. This response could be adapted, promoting the sperm-egg interaction and the fusion of their membranes [169]. A detailed study of this effect would be necessary for mammal species, including the potential role that TRPV1 could exert in this sperm ability.

More recently, several studies examined the regulatory role played by microRNAs (miRNAs) in posttranscriptional regulation of TRPV1. For instance, it was demonstrated that miR-338-3p directly targets TRPV1, increasing mechanical allodynia and thermal hyperalgesia as well as the expression of inflammation-associated genes (COX-2, TNF-α, and IL-6) [80]. Regarding the spermatozoa, dysregulation of miRNAs has been correlated to male infertility [170], allowing to hypothesize a correlation between miRNAs and TRPV1 receptor also in spermatozoa. Furthermore, numerous researchers have correlated the up- or down-regulation of different miRNAs in asthenozoospermic, oligoasthenozoospermic and/or tetrazoospermic patients compared to normozoospermic men, showing a potentially interesting role in the diagnosis of male infertility [121,171,172,173]. However, further research is needed to decipher their exact correlation.

### 5.2. Thermotaxis and TRPV1: Role in Human Spermatozoa

In humans, the presence of the TRPV1 receptor has been demonstrated in both the spermatozoa and the testis [8,99]. In spermatozoa, the expression of this receptor is associated with the sperm’s ability to move toward a temperature gradient. While chemotaxis is considered to exert short-range guidance [174], thermotaxis is chemically defined as the movement toward or away from a thermal stimulus, and it has been studied in mammalian spermatozoa as long-range guidance. It is well accepted that during ovulation in mammals, a difference of temperature is established between the sperm reservoir and the fertilization site, reaching a gap of approximately 2 °C in some species as rabbit [175] or 0.7 °C in swine [176]. Moreover, it has been demonstrated the sperm has the capability to sense small temperature differences, responding even to changes as little as 0.0006 °C in humans [177]. Even if the exact molecular mechanisms for sperm migration by thermotaxis are completely unknown, TRPV1 has been identified as one of the receptors able to identify the temperature fluctuations and guide the spermatozoa across the oviduct toward the fertilization site. Furthermore, the involvement of the distal half of the C-terminal domain in the thermal sensitivity [178] was demonstrated, shedding some light into the function of this protein domain.

The exposure of spermatozoa to a higher temperature in some tracts of the female oviduct (specifically, in the fertilization site) has been shown to influence the membrane structure and fluidity as well [163]. Studies performed in humans showed that non-capacitated spermatozoa are able to migrate toward a gradient of 31–37 °C, demonstrating an amplified response in capacitated spermatozoa. De Toni and coll. (2016) demonstrated thus the role of TRPV1 as a mediator of sperm thermotaxis in humans, related to the calcium-mobilizing properties of TRPV1 and the calcium-dependent increase in motility observed during sperm migration by thermotaxis [8]. Indeed, sperm cells that preferentially move toward a higher temperature gradient have an increased expression of the TRPV1 channel when compared to non-migrating cells. Despite these important results, it is yet to be clarified the exact distribution and localization of TRPV1 in human spermatozoa [179]. From a molecular point of view, the study of TRPV1 function could be of great importance to decipher if the protein is directly involved in thermotaxis or in concomitant processes (i.e., capacitation or acrosome reaction [179]). This will surely bring new insights regarding the role of TRPV1 in sperm function and also contribute to the discovering of new therapeutic strategies to treat male infertility.

### 5.3. Sperm Rheotaxis

Sperm rheotaxis has been suggested as a major guidance and selection force during the sperm migration toward the egg; however, the exact mechanism of action remains unknown, and some of the results obtained over the last few years are contradictory. Earlier experiments performed by Miki and Clapham suggested that this form of guidance may be due to the spiral rotation of the sperm tail caused by the increase in the intracellular Ca^2+^ concentrations, which increases the amplitude of the distal tail excursion and thus the tangential force, producing an effect of sperm hyperactivation. This condition, characterized by asymmetric and larger amplitude flagellar waveform, helps the sperm cells to swim in viscous solutions [139]. Additionally, mechanical forces have been studied on mammalian sperm in relation to the sperm detachment from the oviductal epithelial cells, which may be assisted by a form of mechanical hyperactivation [180,181]. Other than TRPV4, TRPV1 is also recognized as a sensor for hyperosmolarity in some cell types [182], playing a pivotal role in cell homeostasis and ion balance. This faculty to act as an osmolarity sensor is supposed to help in the sperm journey along the oviduct.

Regarding the rheotactic movements induced by the intracellular calcium, later experiments were realized in human sperm by Zhang et coll. (20016) aimed to classify rheotaxis as a passive or active process revealed the absence of differences in flagellar beating and intracellular Ca^2+^ between freely swimming spermatozoa and sperm guided by rheotaxis. While the authors maintain that rheotaxis is a passive process with no flow sensing involved, they hypothesize as well that weak or not perceptible signaling derived from the Ca^2+^ entry into the cell may occur [183], something supported as well by other researchers [139,184] and that may count on CatSper as the channel for the calcium entry [139]. Very recently, Schiffer and collaborators suggested through their experiments in humans and mice that sperm rheotaxis was enabled by passive biomechanical and hydrodynamic processes rather than calcium signaling mediated by CatSper [185]. Even if a role for TRPV1 could be easy to hypothesize due to the potential participation of calcium signaling and their multiple well-known modulators met along the sperm journey, what it is clear here is that further experiments would be necessary to establish the exact mechanism by which this migration force occurs, the participation levels for TRPV1 and the commitment with chemo- and thermotaxis.

## 6. TRPV1 as a Pharmacological Target and Future Perspectives

Studying mice mutants is a useful tool to study numerous pathologies in human and other animal species. Numerous researchers have used the homozygous mutant for TRPV1 in mouse species to carry out their research activity. Trpv1 KO (trpv^−/−^) are viable and with no gene product in dorsal root ganglia and low or absent response to vanilloid compounds, heat, or pH variations. Moreover, they are characterized by a lack of response after capsaicin subcutaneous injection and deficits in thermally evoked pain behavior. Interestingly, TRPV1 KO mice live longer than wild-types, displaying a youthful metabolic profile, a higher tolerance to glucose, better motor coordination, and reduced aging. It was recently demonstrated that TRPV1 is crucial to maintain a constant body temperature, being necessary for thermal homeostasis [186]. Additionally, TRPV1 KO mice are fertile, thus complicating the enlightenment of the relationship between TRPV1 expression and male infertility. However, it is easily explained due to the coexistence of redundant mechanisms as a safety strategy, by which the failure of one of them does not affect the efficiency of the whole process [187]. Thus, to date, it is clear that TRPV1 plays a role in sperm function and capacitation in a mechanism related to the endocannabinoid system [188]. Lewis and collaborators demonstrated a decrease in the TRPV1 binding to CB receptors translated into a loss of function in infertile patients. However, the mRNA and protein levels remained stable, complicating the clarification of these decreases and thus the effect on capacitation and the sperm function [107].

Regarding the pharmacological spectrum, the aim of the TRPV1 study has been focused mainly on the area of analgesics and pain relief by exploiting the agonists/antagonists properties of the great number of these protein modulators. Many compounds have been developed to either activate or inhibit this receptor, but to date, none of them are included in routine clinical practice. However, further experiments are needed with the scope of deciphering the TRPV1 ligands and the interaction and binding sites, as well as the complete structure of the protein within the membrane. Furthermore, the data gathered here bring to light the role of TRPV1 in the process of sperm migration-related specifically to chemo- and thermotaxis guidance mechanisms, which could constitute a new therapeutical tool to be applied for sperm selection in human as in other animal species.

## Figures and Tables

**Figure 1 ijms-22-04306-f001:**
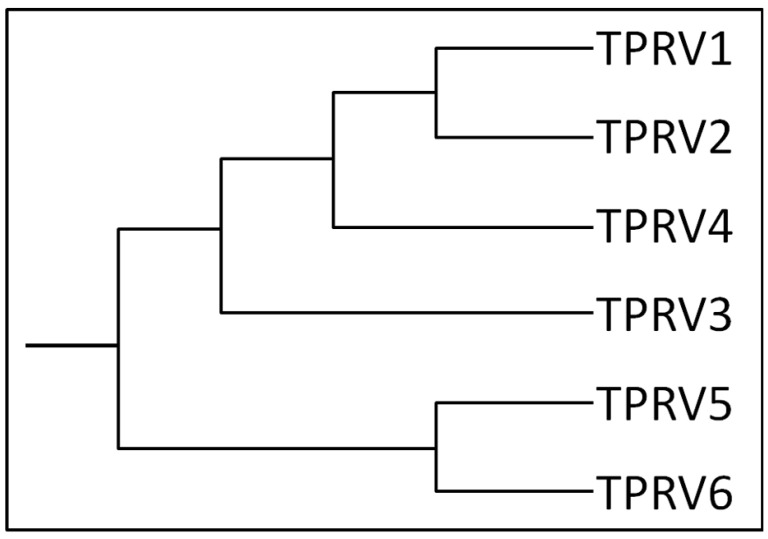
Human TRPV channel subfamily dendrogram. The schema shows a graphical phylogenetic reconstruction of the TRPV subfamily, created from the alignment and phylogenetic reconstruction obtained after using ClustalW.

**Figure 2 ijms-22-04306-f002:**
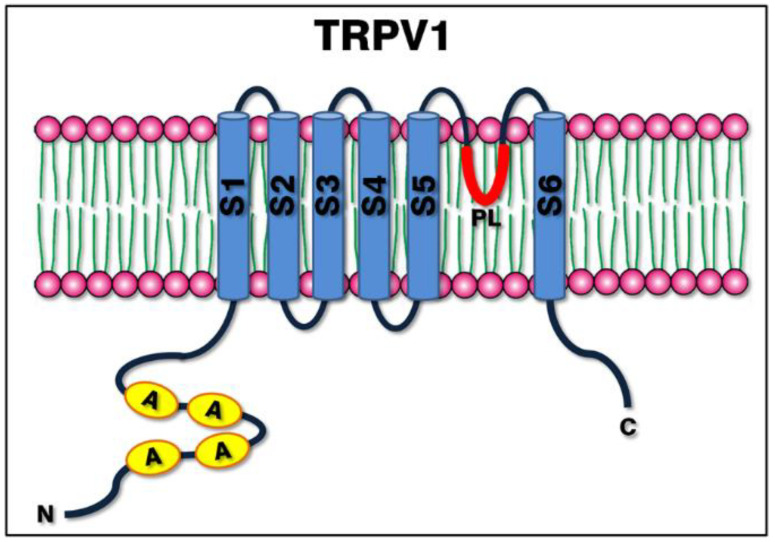
Structure of the TRPV1 channel. The image shows a simplified schema of the protein structure within the plasma membrane, consisting of six transmembrane helices (S1 to S6), a hydrophobic pore loop (PL) between S5 and S6, the N-terminal domain rich in ankyrin repeats and the C-terminal domain.

**Figure 3 ijms-22-04306-f003:**
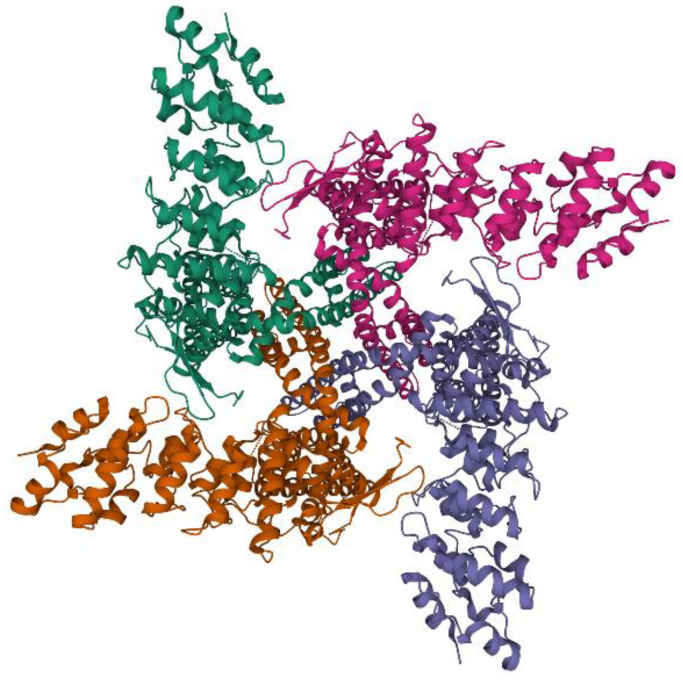
Three-dimensional structure of TRPV1. The image shows a caption of the protein structure obtained by Liao et coll. (2013) using single-particle electron cryo-microscopy [26]. Obtained from https://www.rcsb.org/structure/3J5P (accessed on 20 April 2021), PDB ID 3J5P [29,30].

**Figure 4 ijms-22-04306-f004:**
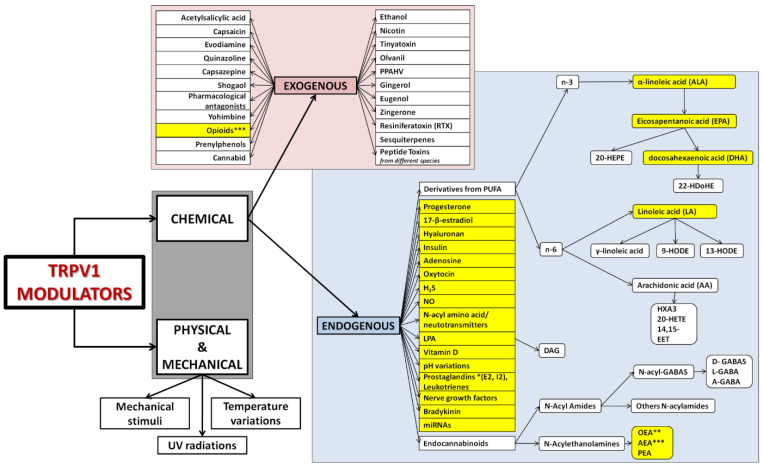
Modulators of TRPV1 function. The schema shows most of the activators and modulators of TRPV1 function investigated to date, divided into two main groups: physical and physical stimuli; and chemical activators. Physical and mechanical modulators include temperature variations (heat or cold) [34], UV radiation [35,36], and mechanical stimuli [37]. The chemical activators group is divided in turn into two subgroups depending on the origin of the activator/modulator: exogenous and endogenous. Exogenous stimuli are capsaicin [38] and its antagonist capsazepine [39], resiniferatoxin (RTX) [37,40], bradykinin [41], yohimbine [42], ethanol [43], evodiamine [44], 17-β-estradiol [45], quinazoline [46], progesterone [47], numerous opioids [48], nicotine [49], hyaluronan [50], insulin [51], tinyatoxin [52], olvanil [53], acetylsalicylic acid [54], eugenol [55], sesquiterpenes [56,57,58], cannabidiol [59], prenylphenols [60], zingerone [61], shogaol [61], PPAHV [62], gingerol [63], numerous pharmacological antagonists [64,65,66,67], and peptide toxins from different species [4,68]. Endogenous activators comprise numerous compounds [1], such as the wide family derived from polyunsaturated fatty acids (PUFAs). According to the position of the first double bond present in their structures, this big family can be divided into n-3 (α-linolenic acid (ALA) and its products eicosapentanoic acid (EPA), docosahexaenoic acid (DHA), 20-hydroxyeicosapentaenoic acid (20-HEPE) and 22-hydroxyeicosapentaenoic acid (22-HDoHE)), and n-6 linoleic acid (LA) and its products γ-linoleic acid, arachidonic acid (AA) [69] (which produces hepoxylin, HXA-3, 14,15-epoxyeicosatrienoic acid, 14,15-EET, and 20-hydroxyeicosatetranoic acid, 20-HETE), 9- and 13-hydroxyoctadecadienoic acids (9-HODE and 13-HODE). Other agonists endogenously secreted are oxytocin [70], adenosine [71], nitric oxide [72], hydrogen sulfide (H_2_S) [73], lysophosphatidic acid (LPA) [74] and its derivative diacylglycerol [75], pH variations [76,77], vitamin D [78], N-acyl amino acids/neurotransmitters [79], miRNAs [80,81], prostaglandins, nerve growth factors and the endocannabinoids family, including N-Acyl amides [82] (N-acyl GABAS [83,84] derived then in D-GABAs, L-GABA and A-GABA) and N-acylethanolamines (NAEs) [85], such as anandamide (AEA) *** [86], oleylethanolamine (OEA) [87] and palmitoylethanolamine (PEA) [88]. * Product from arachidonic acid. ** Derived also from oleic acid, identified as a TRPV1 natural inhibitor [89]. *** Can be exogenous and endogenous. Compounds evidenced in yellow have been found to produce an effect also in spermatozoa, being synthesized by them or present in the oviductal fluid or female reproductive tract.

**Figure 5 ijms-22-04306-f005:**
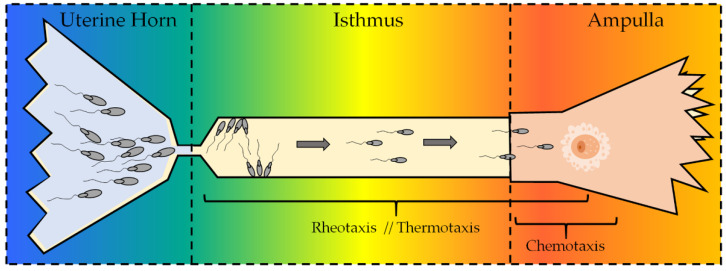
Female oviduct and sperm migration. The diagram shows the female tract of mammals and the three sperm guidance mechanisms, depending on the site in which they have been described to act. While rheotaxis and thermotaxis have been suggested to act mostly in long-range distances, chemotaxis is supposed to take place only in the proximity of the female egg (considering that the exact fertilization site is species-dependent). The colors represent the thermal and chemical gradient (from left to right).

**Table 1 ijms-22-04306-t001:** Functional sperm TRPV1 channels described in different vertebrates.

Animal Group (Species or Strain)	Year of Publication	Reference
Bovine *(Bos taurus)* Swine *(Sus scrofa)* Rodent *(Swiss CD-1 and Sprague-Dawley)* Fish *(Labeo rohita)* Duck *(Anas platyrhynchos)* Human *(Homo sapiens)*	2011 2005 2010, 2009 2013 2020 2016	[98] [14] [100,102] [97] [18] [8]

**Table 2 ijms-22-04306-t002:** Modulators of TRPV1 function found in the female reproductive tract, seminal plasma, or spermatozoa in different mammals.

TRPV1 Modulator	Female Reproductive Tract	Seminal Plasma/Spermatozoa	Animal Group *	Reference
17-β-estradiol	X		Bovine, swine, rabbit, equine	[103,104,105]
Anandamide (AEA)	X	X	Bovine, human, swine, murine	[14,106,107,108]
Arachidonic acid (AA) **		X	Human	[109,110]
Bradykinin	X		Swine	[111]
Docosahexaenoic acid (DHA) **		X	Human	[109,110,112]
Eicosapentanoic acid (EPA) **		X	Human	[109,110]
H_2_S	X	X	Human	[113,114,115]
Hyaluronan	X		All mammals	[116,117]
Insulin		X	Human	[118]
Leukotriene B4	X		Bovine	[119]
Linoleic acid (LA) **		X	Human	[109,110]
miRNAs	X	X	Murine, bovine, human	[120,121,122]
N-acyl amino acids	X	X	Human	[123]
Nerve growth factors		X	Rabbit, bovine	[124,125]
NO	X	X	Bovine, human, swine, mouse, equine	[113,115,126]
Oleoylethanolamine (OEA)	X		Bovine	[106]
Opioids		X	Human	[127]
Oxytocin		X	Human	[128]
Palmitoylethanolamine (PEA)	X		Bovine	[106]
pH variations	X		Human	[129]
Progesterone	X		Bovine, swine, rabbit, equine	[103,104,105]
Prostaglandins	X		Bovine	[126]
Vitamin D	X	X	Human	[130,131,132]
α-linolenic acid (ALA) **		X	Human	[109,110]

* Example (but not limited to) of the animal species where the modulator was found. ** Component of the sperm phospholipid membrane.

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
