# Peer review of "Role and Modulation of TRPV1 in Mammalian Spermatozoa: An Updated Review"

_ijms, 2021, doi:10.3390/ijms22094306_

Round 1
Reviewer 1 Report
My comments to the authors are included in the file (Word document) provided.

Author Response
Authors Response:
First of all, we would like to sincerely thank the Reviewer 1 for his/her exhaustive analysis of our manuscript, which we have really appreciated. We have addressed every general and specific comments and suggestions made by the Reviewer. This submission is then accompanied by a “clean manuscript” and a “highlighted modifications” version, the latest including all the modifications clearly highlighted in yellow. Once all the corrections were made, the clean version of the manuscript was subjected again to a revision by a native English speaker.
We consider that the quality of our manuscript has considerably increased thanks to the Rev 1 comments, and we hope it could be re-considered for publication in J of Mol Sciences.
Thank you very much for contributing to improve this work.
Response to specific comments:
Line 190: “… some new compounds.” Such as?
Response: Unfortunately, we don’t have this information. In the cited manuscript, compounds were named “Compound 1, compound 2 and compound 3” by the Authors (maybe because of patent reasons). However, we specified this issue in the text (now line 189 of the clean manuscript version).
Line 408: I don’t think ‘dysregulation’ is the appropriate term here; please reword.
Response: we use this term because it is used among the researchers working within that field. Since we did not find a better term, we maintained it here. We apologize for the inconveniences.

Reviewer 2 Report
Based on the literature review, the authors compiled the latest information on the TRPV1 receptor, which is an ion channel involved in calcium homeostasis. It is found among others in sperm cells and plays an important role during their capacitation in the female reproductive tract. Updating information on this protein is necessary, as TRPV1 participates in very important physiological processes of sperm that take place during their post-ejaculatory maturation (thermotaxis and chemotaxis) and also during fertilization (acrosomal reaction). Undeniably, these processes depend on the concentration of calcium. Their impairment can lead to idiopathic infertility.
In the first part of the manuscript, the authors paid special attention to the subfamily of TRPV1 belonging to the family TRP. They comprehensively described the presence of TRPV1 in various cells and tissues, and its protein structure. The authors went on to analyze the function, modulators and regulation mechanism of TRPV1 channel exhaustively. Information contained in the manuscript has been visualized using figures and diagrams, that in an educational and accessible way allow the reader to understand the unusual role of TRPV1 protein for cell function in biological systems and under specific physiological and pathological conditions.
The rest of the manuscript focuses on the role of TRPV1 in sperm. The factors found in sperm, seminal plasma and in the female genital tract secretions that affect the function of TRPV1 have been exhaustively listed. The relationship between TRPV1 function and chemotaxis and thermotaxis occurring in the fallopian tube has been revealed. In addition, the role of the endocannabinoid system in sperm chemotaxis has been considered. This part of the manuscript contains the necessary tables and figures to organize the knowledge contained in the text. The last part of the manuscript is devoted to the use of the TRPV1 receptor as a pharmacological target, which could help in the treatment of infertility in the future.
I would like to emphasize that the authors have extensive knowledge of the molecular basis of capacitation, which aims to select those male gametes that are the most biologically attractive. The processes taking place during the long migration of sperm through the female genital tract are complicated, multi-stage, synchronized in time, regulated by many factors of female and male origin, and take place in a specific sequence depending on the section of the female genital tract. Understanding and describing them in an accessible way requires the authors of comprehensive knowledge at the molecular, cellular and physiological levels. In the opinion of the reviewer, the authors achieved the assumed goal.
Author Response
Authors Response:
We would like to sincerely thank the Reviewer 2 for his/her exhaustive analysis of our manuscript, demonstrating a high expertise in the field, and his/her positive evaluation, which we really appreciated. We would just like to inform that the English language was revised and some modifications were carried out, following the suggestions made from Reviewer 1.
Thank you very much.

Round 2
Reviewer 1 Report
I have attached a Word document for the authors.

Author Response
Dear Reviewer 1,
Thank you very much for your time and your kind suggestions and comments. One more time, we have taken them into consideration, and all the modifications were done following your specific comments. Please, note that some of them were referred to the first version of the manuscript, not to the second submission. Thus, they had been already corrected during the first revision (and submitted with the second submission). References were curated, added and changed when needed, and the manuscript was again revised by a native English speaker.
We would like thank you again for your considerable help in improving this manuscript, which we hope it could be reconsidered for publication now.
Kindest regards,
The Authors
